

# Identification of a circadian gene signature that predicts overall survival in lung adenocarcinoma

Xinliang Gao[1], Mingbo Tang[1], Suyan Tian[2], Jialin Li[1] and Wei Liu[1]

[1] Department of Thoracic Surgery, The First Hospital of Jilin University, Changchun, Jilin Province, China
[2] Division of Clinical Research, The First Hospital of Jilin University, Changchun, Jilin Province, China

## ABSTRACT

**Background.** Lung adenocarcinoma (LUAD) is one of the most common subtypes of lung cancer which is the leading cause of death in cancer patients. Circadian clock disruption has been listed as a likely carcinogen. However, whether the expression of circadian genes affects overall survival (OS) in LUAD patients remains unknown. In this article, we identified a circadian gene signature to predict overall survival in LUAD.
**Methods.** RNA sequencing (HTSeq-FPKM) data and clinical characteristics were obtained for a cohort of LUAD patients from The Cancer Genome Atlas (TCGA). A multigene signature based on differentially expressed circadian clock-related genes was generated for the prediction of OS using Least Absolute Shrinkage and Selection Operator (LASSO)-penalized Cox regression analysis, and externally validated using the GSE72094 dataset from the GEO database.
**Results.** Five differentially expressed genes (DEGs) were identified to be significantly associated with OS using univariate Cox proportional regression analysis ($P < 0.05$). Patients classified as high risk based on these five DEGs had significantly lower OS than those classified as low risk in both the TGCA cohort and GSE72094 dataset ($P < 0.001$). Multivariate Cox regression analysis revealed that the five-gene-signature based risk score was an independent predictor of OS (hazard ratio > 1, $P < 0.001$). Receiver operating characteristic (ROC) curves confirmed its prognostic value. Gene set enrichment analysis (GSEA) showed that Kyoto Encyclopedia of Genes and Genomes (KEGG) pathways related to cell proliferation, gene damage repair, proteasomes, and immune and autoimmune diseases were significantly enriched.
**Conclusion.** A novel circadian gene signature for OS in LUAD was found to be predictive in both the derivation and validation cohorts. Targeting circadian genes is a potential therapeutic option in LUAD.

## INTRODUCTION

Lung cancer is a leading cause of death in the world (*Bray et al., 2018*). The estimated 5-year survival rate is only 19% (*Siegel, Miller & Jemal, 2019*). In 2019, there were 228,150 new diagnoses of cancers of the lung and bronchus in the United States. Primary lung cancer is divided into two main types: small-cell lung carcinoma and non-small cell lung carcinoma (NSCLC). The latter is further classified into different subtypes according to the

Corresponding author
Wei Liu, l_w01@jlu.edu.cn

histological origin, such as lung adenocarcinoma (LUAD), squamous cell carcinoma, or large cell carcinoma. Among these, LUAD is the most prevalent subtype, with an increasing incident in recent years (*Cheng et al., 2016*). The prognosis of LUAD is improving due to advances in molecular targeted treatment and immunotherapy (*Hirsch et al., 2016*; *Peters et al., 2019*). However, accurate prognosis prediction models for LUAD are still lacking.

The circadian clock is a molecular time-keeping system that is evolutionarily conserved. It is vital for the maintenance of physiologic homeostasis and normal function in all organisms. It coordinates a variety of biological processes and behaviors (*Fu & Kettner, 2013*; *Panda et al., 2002*). In the suprachiasmatic nucleus (SCN) of the hypothalamus, a central clock maintains the daily rhythms in the body by neural and humoral communication with peripheral clocks located in peripheral tissues and regulates bodily functions such as sleep/wake cycles and the secretion of many hormones. Disruption of the circadian clock has been listed as a likely carcinogen by the World Health Organization based on both population and laboratory-based findings (*Lunn et al., 2017*; *Straif et al., 2007*), which raised the interest in research on the relationship between circadian genes and tumor development. Some circadian genes have been demonstrated to control the occurrence and development of NSCLC (*Qiu et al., 2019*). However, the association between circadian genes and prognosis in patients with LUAD remains to be elucidated.

The present study aims to explore the prognostic role of circadian genes in patients with LUAD using The Cancer Genome Atlas (TCGA) data obtained from the NCI Genomic Data Commons, which includes the clinical characteristics and mRNA expression profiles of tumor and tumor-adjacent normal tissues. A prognostic multigene signature will be established using differentially expressed circadian clock genes and then validated with the GSE72094 dataset extracted from the Gene Expression Omnibus (GEO) database. Underlying molecular mechanisms were investigated by performing a Gene set enrichment analysis (GSEA) with Kyoto Encyclopedia of Genes and Genomes (KEGG) pathways.

## MATERIALS & METHODS

### Data collection

The clinical characteristics and RNA sequencing data (HTSeq-FPKM) of 515 patients with LUAD were retrieved from the NCI Genomic Data Commons (https://portal.gdc.cancer.gov/repository). These 515 patients provided 535 samples from LUAD tumor tissue and 59 samples from adjacent normal tissue. Among the patients, 500 had complete RNA sequencing data and 469 had both complete sequencing data and complete clinical information.

The differential expression of the following 14 core genes of the circadian clock according to previous literature was analyzed: *Period 1* (*PER1*), *PER2*, *PER3*, Cryptochrome Circadian *Regulator 1* (*CRY1*), *CRY2*, *Circadian Locomotor Output Cycles Kaput* (*CLOCK*), *Aryl Hydrocarbon Receptor Nuclear Translocator Like* (*ARNTL*), *Timeless Circadian Regulator* (*TIMELESS*), *Neuronal PAS Domain Protein 2* (*NPAS2*), *Nuclear Receptor Subfamily 1 Group D Member 1* (*NR1D1*), *NR1D2*, *Basic Helix-Loop-Helix Family Member E40* (*BHLHE40*), *BHLHE41*, and *RAR-Related Orphan Receptor A* (*RORA*) (*Chen et al., 2020*;
*Cox & Takahashi, 2019*; *Mocellin et al., 2018*; *Shafi & Knudsen, 2019*; *Yu et al., 2019*). The validation dataset was obtained from the GSE72094 dataset in the GEO database (https://www.ncbi.nlm.nih.gov/geo/) and included microarray and clinical data for 443 LUAD tumor samples (*Schabath et al., 2016*). The normalized count data were downloaded. The data cut-off date was September 10, 2020. Patients with no follow-up data or information on the expression of circadian genes were excluded.

The TCGA and GEO databases are public data repositories and therefore, ethical approval for this study was not required. This study followed the polices and guidelines for data access and publication specified by the TCGA and GEO databases.

## Prognostic validity of the gene signature

Differentially expressed genes (DEGs) involved in the circadian clock were analyzed in the tumor and tumor-adjacent normal tissues of LUAD patients from the TCGA cohort using the "limma" package in R (false discovery rate (FDR) <0.05). Univariate Cox regression analysis was used to identify circadian genes related to overall survival (OS). A gene signature for the prediction of OS was constructed with the DEGs for the circadian clock using Least Absolute Shrinkage and Selection Operator (LASSO)-penalized Cox regression analysis and the "glmnet" package in R. DEGs served as independent variables, and OS as the response variable.

A risk score based on the expression of identified candidate genes was calculated for each patient according to the following formula: score = sum (normalized gene expression level × regression coefficient). Patients were classified as either high- or low-risk using the median score as the cut-off value. The survival analysis of different risk groups was determined with the "survminer" R package. In order to validate the performance of the signature, we used the principal components analysis (PCA) and t-distributed stochastic neighbor embedding (t-SNE) to analyze dimensionality reduction. The "prcomp" function in the R "stats" package was used to carry out the PCA. The data distribution for high-risk and low-risk patients was also mapped using t-SNE and the "Rtsne" package in R. The predictive value of the gene signature was evaluated with time-dependent Receiver operating characteristics (ROC) curve analysis using the "timeROC" package in R. The associations between the risk score, clinical characteristics (gender, age, smoking history, and stage), and OS were assessed with univariate and multivariate Cox regression analyses.

## Functional enrichment analysis

The DEGs between the high- and low-risk groups in the TCGA LUAD cohort were identified using the "limma" R package again. GSEA of these DEGs was carried out with KEGG pathways ($|\log2$ fold change$| \geq 1$, FDR <0.05). Both a nominal $P$-value <0.05 and FDR $q$-value <0.05 were considered statistically significant.

## Statistical analysis

All statistical analyses were conducted with R software (Version 3.5.3) and SPSS software (Version 25.0). Gene expression was compared using the two-tailed Student's $t$-test and proportions were compared using the Chi-squared test. The Kaplan–Meier method and the log-rank test were used to assess the differences in OS between high and low-risk patients.

**Table 1  Demographical and clinical characteristics.**

| | TCGA LUAD | GSE72094 | P value |
|---|---|---|---|
| No. of patients | 469 | 328 | |
| Age (median, range) | 65.1 (33–88) | 69.7 (41–89) | $P < 0.01$ |
| Gender (%) | 257 (54.8%) | 172 (52.4%) | $P = 0.559$ |
| Female | 212 (45.2%) | 156 (47.6%) | |
| Male | | | |
| TNM Stage | 257 (54.8%) | 218 (66.5%) | $P = 0.007$ |
| I | 112 (23.9%) | 53 (16.2%) | |
| II | 75 (16.0%) | 46 (14.0%) | |
| III | 25 (5.3%) | 11 (3.4%) | |
| IV | | | |
| Smoking history | 69 (14.7%) | 30 (9.1%) | $P = 0.025$ |
| Non-smoker | 400 (85.3%) | 298 (90.9%) | |
| Smoker | | | |
| Median OS (days) | 629 | 842 | $P = 0.034$ |

Univariate and multivariate Cox regression analyses were used to identify independent predictors of OS. $P < 0.05$ (two tailed) was considered statistically significant.

# RESULTS

## Clinical and demographic characteristics

Two patient cohorts with available data on OS and the RNA expression of circadian clock genes were used to create the prognostic model. The derivation cohort consisted of 500 patients with LUAD and complete RNA sequencing data from the TCGA database while the validation cohort consisted of 398 patients with LUAD from the GSE72094 dataset. Among these patients, 469 patients from TCGA and 328 patients from GSE72094 who not only had complete RNA sequencing data, but also complete clinical data including OS, age, gender, smoking history, and tumor stage, were included in the univariate and multivariate COX analyses. The validation cohort had higher age, lower TNM stage, more smokers, and a higher median OS compared to the derivation cohort. The baseline demographic and clinical characteristics of the included patients are summarized in Table 1.

## Identification of DEGs related to circadian clock in the TGCA LUAD cohort

In the TCGA LUAD cohort, 9/14 circadian genes were found to be differentially expressed between tumor and tumor-adjacent normal tissues. Five candidate genes were identified to be significantly associated with OS using univariate Cox proportional regression analysis (Figs. 1A–1B). The clustering of the 5 candidate genes are shown with a heatmap in Fig. 1C.

## Generation of a prognostic signature in the TGCA LUAD cohort

The 5 identified candidate genes were incorporated into a five-gene-signature based prognostic model using LASSO Cox regression analysis. According to risk scores calculated using the expression levels of these 5 genes, half of the patients were classified as high-risk ($n = 250$) and the other half as low-risk ($n = 250$) (Fig. 2A). The chance of survival was

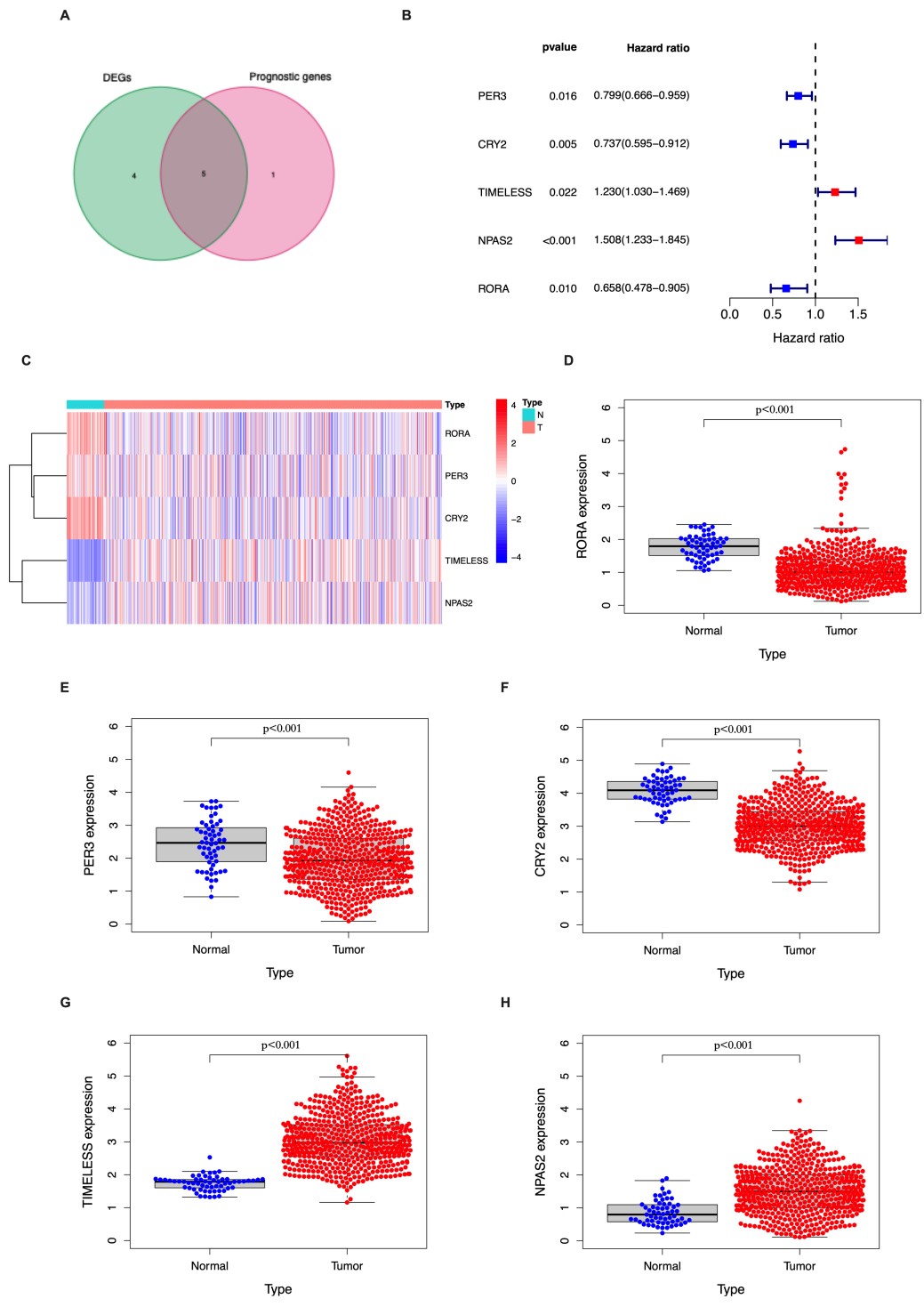

**Figure 1  Identification of the candidate genes involved in the circadian cycle in the TCGA cohort.** (A) Venn diagram of DEGs and prognostic genes that correlate with OS in tumor and tumor-adjacent normal tissue. (B) Forest plots of the five genes that overlap between DEGs and prognostic genes that relate to OS on univariate Cox regression analysis. (C) The mRNA heatmap of five candidate genes. (D–H) The expression of five candidate genes in tumor and normal tissue.

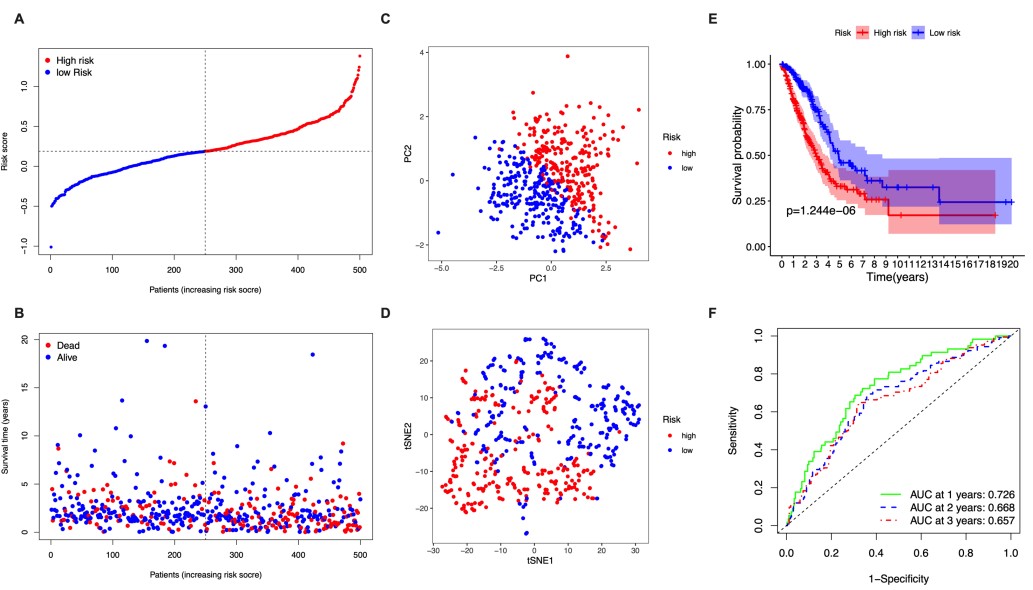

**Figure 2** **Analysis of the prognostic value of the five-gene-signature based risk score in the TCGA cohort.** (A) The distribution of the risk scores in the TCGA cohort. (B) The distributions of OS status, OS time and risk score in the TCGA cohort. (C) PCA plot of the TCGA cohort. (D) t-SNE analysis of the TCGA cohort. (E) Kaplan–Meier curves for the OS of patients in the high- respective low-risk group in the TCGA cohort. (F) AUC of time-dependent ROC curves that confirm the prognostic performance of the risk score in the TCGA cohort.

lower and the survival time was shorter in the high-risk group than in the low-risk group (Fig. 2B). PCA and t-SNE analysis showed discernible dimensions between high-risk and low-risk patients (Figs. 2C–2D). Kaplan–Meier survival curves confirmed that OS was significantly worse in high-risk than in low-risk patients (Fig. 2E, $P < 0.001$). The predictive performance of the five-gene-signature based risk score for OS was evaluated using time-dependent ROC curves. The area under the curve (AUC) values were: 1 year, 0.726; 2 years, 0.668; and 3 years, 0.657 (Fig. 2F).

## Validation of the five-gene-signature based prognostic model

The stringency of the model developed using the TGCA LUAD cohort was validated in the GSE72094 dataset. Risk scores were calculated for all patients based on the expression levels of the 5 identified candidate genes and patients were classified as high-risk or low-risk accordingly (Fig. 3A). The high-risk group had a significantly higher chance of death and lower OS time (Fig. 3B). PCA and t-SNE analysis showed discernible dimensions between high- and low-risk patients (Figs. 3C–3D). Kaplan–Meier survival curves confirmed that OS was significantly worse in high-risk patients (Fig. 3E, $P < 0.001$). The AUC values were: 1 year, 0.621; 2 years, 0.657; and 3 years, 0.642 (Fig. 3F).

## Prognostic value of the five-gene-signature based risk score

Univariate and multivariate Cox regression analyses were conducted to determine whether the five-gene-signature based risk score was an independent predictor of OS (Table 2). The derivation cohort consisted of 469 patients from the TCGA LUAD cohort; and

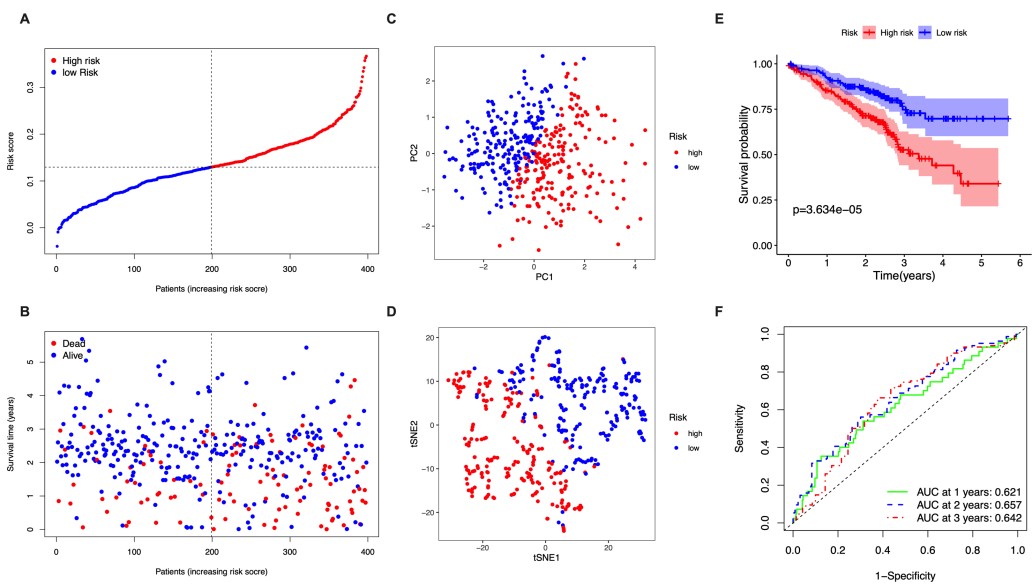

**Figure 3** **Validation of the five-gene-signature based risk score in the GSE72094 dataset.** (A) The distribution of the risk scores in the GSE72094 dataset. (B) The distributions of OS status, OS time and risk score in the GSE72094 dataset. (C) PCA plot of the GSE72094 dataset. (D) t-SNE analysis of the GSE72094 dataset. (E) Kaplan–Meier curves for the OS of patients in the high- respective low-risk group in the GSE72094 dataset. (F) AUC of time-dependent ROC curves that confirm the prognostic performance of the risk score in the GSE72094 dataset.

**Table 2** **Risk factors affecting OS in the TCGA LUAD cohort and GSE72094.**

| Factors | TCGA LUAD | | | | | | GSE72094 | | | | | |
|---|---|---|---|---|---|---|---|---|---|---|---|---|
| | Univariate | | | Multivariate | | | Univariate | | | Multivariate | | |
| | HR | 95% CI | P | HR | 95% CI | P | HR | 95% CI | P | HR | 95% CI | P |
| Age | 1.01 | 0.99–1.02 | 0.302 | 1.02 | 1.00–1.03 | 0.026 | 1.01 | 0.98–1.03 | 0.533 | 1.00 | 0.98–1.02 | 0.973 |
| Gender | 1.12 | 0.83–1.52 | 0.458 | 1.09 | 0.79–1.49 | 0.606 | 1.99 | 1.30–3.05 | 0.002 | 2.32 | 1.49–3.62 | 0.000 |
| Stage | 1.61 | 1.39–1.85 | 0.000 | 1.59 | 1.37–1.84 | 0.000 | 1.70 | 1.40–2.07 | 0.000 | 1.86 | 1.52–2.29 | 0.000 |
| Smoking | 0.91 | 0.60–1.38 | 0.655 | 0.81 | 0.53–1.26 | 0.351 | 1.31 | 0.57–3.01 | 0.523 | 0.87 | 0.37–2.03 | 0.750 |
| Risk score | 3.37 | 2.20–5.17 | 0.000 | 3.53 | 2.26–5.49 | 0.000 | 2.16 | 1.39–3.36 | 0.001 | 2.20 | 1.41–3.41 | 0.000 |

the validation cohort consisted of 328 patients from the GSE72094 dataset. Univariate regression analysis revealed that the risk score was significantly associated with OS in both the TGCA LUAD cohort and the GSE72094 dataset (TGCA LUAD cohort: hazard ratio (HR) = 3.373, 95% confidence interval (CI) = 2.199−5.174, $P < 0.001$; GSE72094 dataset: HR = 2.163 95% CI = 1.393−3.358, $P < 0.001$). The risk score was found to be an independent predictor of OS even after correcting for confounders in multivariate Cox regression analysis (TGCA LUAD cohort: HR = 3.522, 95% CI = 2.260−5.487, $P < 0.001$; GSE72094 dataset: HR = 2.195, 95% CI = 1.411−3.415, $P < 0.001$; Figs. 4A–4B).

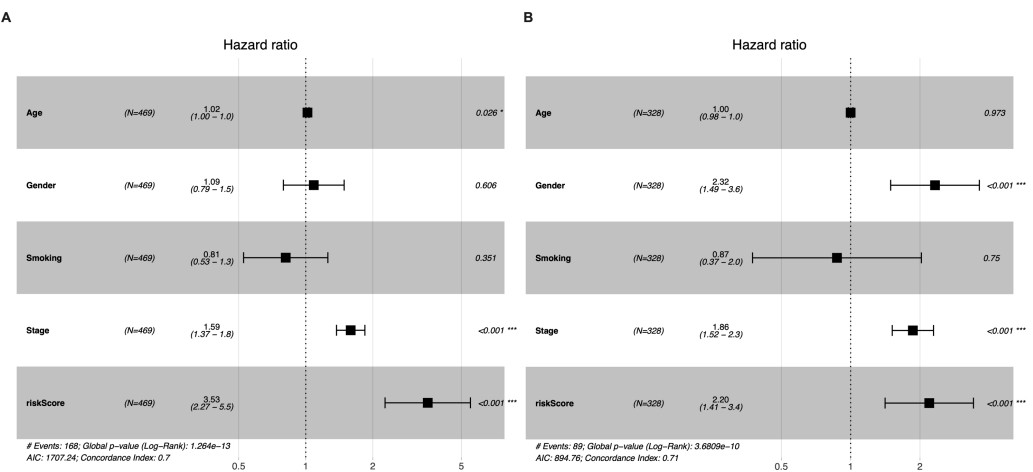

**Figure 4** Multivariate Cox regression analyses of factors affecting OS in the TCGA LUAD cohort (A) and the GSE72094 dataset (B).

## Enrichment analysis in the TGCA LUAD cohort

Genes that were differentially expressed in the high- respective low-risk groups were subjected to GSEA for KEGG pathways (Table 3). The results showed that tumorigenesis pathways related to pyrimidine metabolism, cell cycle, proteasome, base excision repair, homologous recombination, and DNA replication were enriched (Fig. 5).

## DISCUSSION

Genes of the circadian clock are often abnormally expressed in tumor tissues and may play an important role in tumorigenesis (*Kelleher, Rao & Maguire, 2014*; *Kettner, Katchy & Fu, 2014*). The present study identified 9 DEGs between tumor and tumor-adjacent normal tissues among the 14 circadian genes. The genes *PER3*, *CRY2*, *TIMELESS*, *NPAS2*, and *RORA* were found to be correlated with OS. These results suggest that circadian clock genes may affect the survival outcome in LUAD and that a signature based on the expression of these genes may predict OS and may be an independent prognostic factor.

The PER family is generally considered to have a tumor suppressor effect, and the mechanisms behind the tumor suppressing effects of *PER1* and *PER2* are clear (*Gery et al., 2006*; *Wood et al., 2008*). *PER3* has been confirmed to affect the susceptibility and prognosis of lung cancer through expression changes, methylation, and single nucleotide polymorphisms (SNPs) (*Chu et al., 2018*; *Couto et al., 2014*; *Liu et al., 2014*). However, the exact mechanism for the *PER3* inhibition of tumors is not yet clear. The study by Jun-Sub et al. showed that *PER3* is required for *checkpoint kinase 2* (*CHK2*) activation in human cells, which highlighted its potential role in cell cycle arrest and DNA damage repair (*Im et al., 2010*). Previous studies have linked the circadian clock gene *CRY2* to the occurrence and development of many tumors (*Hasakova et al., 2018*; *Lesicka et al., 2018*; *Relles et al., 2013*; *Tokunaga et al., 2008*). As a transcriptional suppressor, *CRY2* functions as an important regulator of cell cycle, proliferation, DNA damage checkpoint control, and DNA

**Table 3  GSEA of DEGs between high-risk group and low-risk group with KEGG pathways.**

| KEGG pathway | NES | NOM p-val | FDR q-val |
|---|---|---|---|
| PYRIMIDINE_METABOLISM | 2.17 | 0.000 | 0.000 |
| CELL_CYCLE | 2.11 | 0.000 | 0.003 |
| SPLICEOSOME | 2.08 | 0.000 | 0.005 |
| PROTEASOME | 2.05 | 0.000 | 0.006 |
| BASE_EXCISION_REPAIR | 2.02 | 0.000 | 0.009 |
| HOMOLOGOUS_RECOMBINATION | 1.97 | 0.000 | 0.015 |
| PATHOGENIC_ESCHERICHIA_COLI_INFECTION | 1.97 | 0.000 | 0.025 |
| DNA_REPLICATION | 1.91 | 0.000 | 0.025 |
| GLYCOSPHINGOLIPID_BIOSYNTHESIS_LACTO_AND_NEOLACTO_SERIES | 1.89 | 0.000 | 0.030 |
| PENTOSE_PHOSPHATE_PATHWAY | 1.88 | 0.006 | 0.030 |
| THYROID_CANCER | 1.87 | 0.002 | 0.032 |
| INTESTINAL_IMMUNE_NETWORK_FOR_IGA_PRODUCTION | −2.11 | 0.000 | 0.011 |
| HEMATOPOIETIC_CELL_LINEAGE | −2.04 | 0.002 | 0.018 |
| ASTHMA | −1.95 | 0.002 | 0.039 |
| AUTOIMMUNE_THYROID_DISEASE | −1.95 | 0.006 | 0.030 |
| CELL_ADHESION_MOLECULES_CAMS | −1.95 | 0.002 | 0.025 |
| PRIMARY_BILE_ACID_BIOSYNTHESIS | −1.91 | 0.000 | 0.032 |
| GLYCOSPHINGOLIPID_BIOSYNTHESIS_GANGLIO_SERIES | −1.90 | 0.004 | 0.029 |
| ALLOGRAFT_REJECTION | −1.89 | 0.010 | 0.031 |
| TYPE_I_DIABETES_MELLITUS | −1.88 | 0.014 | 0.031 |

repair (*Hoffman et al., 2010*). *CRY2* acts as a tumor suppressor gene. It can limit tumor formation by increasing c-MYC turnover (*Huber et al., 2016*), or increase the elimination of premalignant and malignant cells through the activation of p53-independent apoptosis pathways (*Lee & Sancar, 2011*). The circadian genes *NPAS2* and *TIMELESS*, on the other hand, are both correlated with poor OS. A recent study has shown that upregulated *NPAS2* promoted the survival of hepatocellular carcinoma cells through the upregulation of *cell division cycle 25 A* (*CDC25A*) and inhibition of mitochondria-dependent intrinsic apoptosis (*Yuan et al., 2017*). Knockout or inhibition of *TIMELESS* can lead to cell cycle stagnation and subsequent apoptosis, which limits the growth of liver cancer cells (*Elgohary et al., 2015*). The circadian gene *RORA* was found to be downregulated in LUAD tissue and negatively correlated with LUAD prognosis in this study. *RORA* is a versatile gene. Besides the circadian clock, it is also a well-known regulator of inflammation and lipid metabolism. Moreover, recent studies have suggested that *RORA* may also play a role in the progression and prognosis of colon cancer and breast cancer (*Lee et al., 2010*). The recruitment of *RORA* can induce the expression of the tumor suppressor genes *F-box/WD repeat-containing protein 7* (*FBXW7*), *Semaphorin 3F* (*SEMA3F*), and *P21*, leading to apoptosis and suppression of tumor cell proliferation (*Wang et al., 2017*).

Results from the enrichment analysis revealed that metabolic pathways related to the substrates of DNA synthesis (pyrimidine metabolism and pentose phosphate pathway) were enriched in the high-risk group, as well as pathways regulating cell cycle and DNA replication. Increasing evidence suggests a regulatory effect of circadian genes on cellular
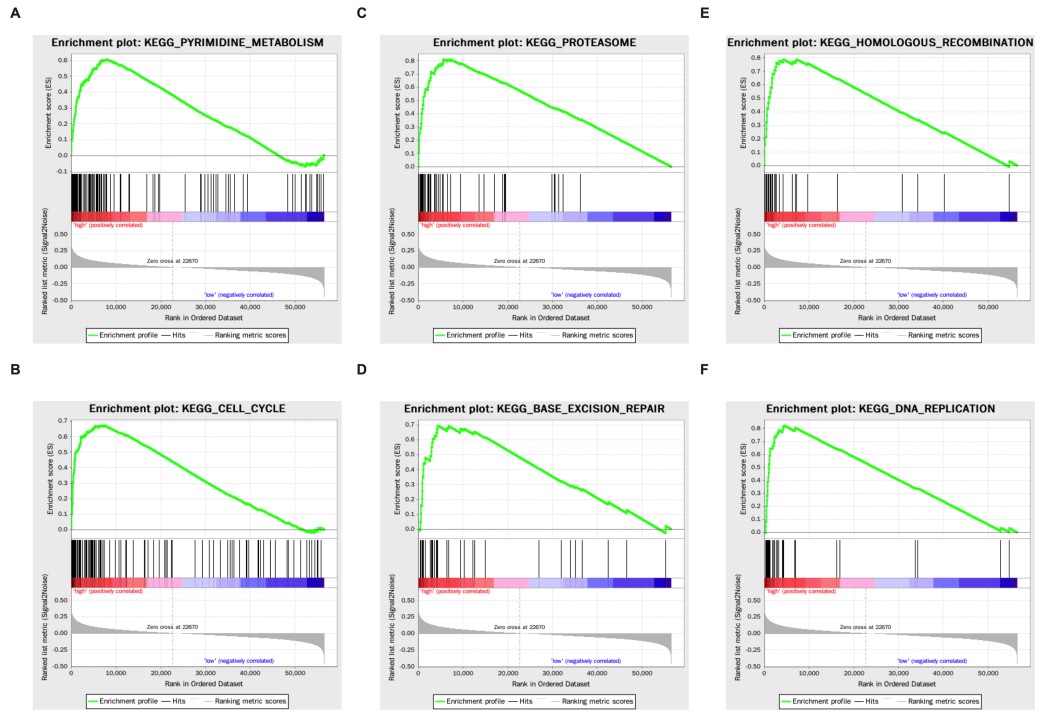

**Figure 5  The KEGG enrichment plots of tumorigenesis pathways.** (A) Pyrimidine metabolism. (B) Cell cycle. (C) Proteasome. (D) Base excision repair. (E) Homologous recombination. (F) DNA replication.

proliferation (*Chakrabarti & Michor, 2020*), and their involvement in the proliferation of a variety of tumor cells (*Abreu et al., 2018*; *Wang et al., 2016*; *Yu et al., 2018*). A recent study on lung cancer demonstrated that the loss of the central clock components led to increased c-MYC expression, which enhanced proliferation (*Papagiannakopoulos et al., 2016*). Base excision repair and homologous recombination pathways were also found to be enriched in the high-risk group, which may indicate that the circadian clock disorder affects the repair of gene damage to influence the survival of malignant tumors. Both *CRY* and *TIMELESS* are known to be involved in DNA damage repair. Tae et al. found that *CRY* s are related to the nucleotide excision repair gene *XPA* (*Kang, Reardon & Sancar, 2011*). *TIMELESS* can modulate *CHK1* and *serine/threonine-protein kinase* (*ATR*) downstream of single-strand DNA breaks and activate *CHK2* via *ATM* modulation downstream of double strand breaks (*Yang, Wood & Hrushesky, 2010*). The proteasome pathway was enriched in the high-risk group. Recent studies have also confirmed that some ubiquitin ligases participate in the degradation of core circadian clock genes through the ubiquitin-proteasome pathway, thereby controlling the biological functions of cells, including cell senescence (*Chen et al., 2018*; *Ullah et al., 2020*). This cross-talk between circadian clock genes and the ubiquitin-proteasome pathway may be related to the prognosis of LUAD. Some immune and autoimmune disease pathways were enriched in the low-risk group. This shows that the disturbance of the circadian clock is accompanied by alterations in the function of the immune system (*Aiello et al., 2020*), which may be related to the

occurrence, development, and prognosis of LUAD. Wu and his colleagues have shown that abnormal circadian genes contribute to T cell exhaustion and global upregulation of immune inhibitory molecules, such as programmed death-ligand 1 (PD-L1) and cytotoxic T-lymphocyte antigen (CTLA)-4, which promote tumor development (*Wu et al., 2019*).

There are several limitations to this study. Firstly, the present study is a retrospective study with data from publicly available databases. This makes the study more prone to selection bias and it is also impossible to draw conclusions regarding cause–effect. Experimental studies should be conducted to understand the mechanisms behind the role of the circadian genes. Secondly, using tumor-adjacent normal tissue as a control has the advantages of minimizing biological variation, but one cannot be sure if the seemingly "normal" tissue adjacent to a tumor is truly "normal". Thirdly, while there might be many other genes that are important in LUAD, we only focused on 14 core genes of the circadian clock. It is possible that other more important genes were excluded from the design.

## CONCLUSIONS

In summary, we constructed a novel five-gene signature with genes involved in the circadian clock to predict the prognosis of LUAD. The signature could successfully separate LUAD patients with a low risk of non-survival from those with a high risk in both the derivation and validation cohorts. The underlying molecular mechanisms between circadian genes and tumor proliferation, DNA repair, ubiquitin-proteasome pathway, and immunity in LUAD remain poorly understood. and warrant further investigation. Circadian genes might be potential targets for future cancer therapy.

## ACKNOWLEDGEMENTS

This manuscript has been edited and proofread by Medjaden Bioscience Limited.

### Funding

This work was supported by the Jilin Province Scientific and Technological Department, International Scientific and Technological Cooperation Project (20190701043GH), Wu Jieping Medical Foundation (No. 320.6750.19092-1), and the Development Center for Medical Science & Technology, National Health Commission of the People's Republic of China (No. WA2020RW18). The funders had no role in study design, data collection and analysis, decision to publish, or preparation of the manuscript.

### Grant Disclosures

The following grant information was disclosed by the authors:
Jilin Province Scientific and Technological Department, International Scientific and Technological Cooperation Project: 20190701043GH.
Wu Jieping Medical Foundation: 320.6750.19092-1.
Development Center for Medical Science & Technology,National Health Commission of the People's Republic of China: WA2020RW18.

## Competing Interests

The authors declare there are no competing interests.

## Author Contributions

- Xinliang Gao conceived and designed the experiments, performed the experiments, analyzed the data, prepared figures and/or tables, authored or reviewed drafts of the paper, and approved the final draft.
- Mingbo Tang and Jialin Li performed the experiments, prepared figures and/or tables, and approved the final draft.
- Suyan Tian conceived and designed the experiments, analyzed the data, authored or reviewed drafts of the paper, and approved the final draft.
- Wei Liu conceived and designed the experiments, authored or reviewed drafts of the paper, and approved the final draft.

## Data Availability

Data is available at TCGA-LUAD (https://portal.gdc.cancer.gov/repository), and at NCBI GEO: GSE72094.

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
