# Peer review of "Identification of a circadian gene signature that predicts overall survival in lung adenocarcinoma"

_PeerJ, doi:10.7717/peerj.11733_

## Round 0.1 · original submission · Major Revisions

Your manuscript was considered novel, interesting and valuable, but there was a couple of issues raised by the reviewers. The first concern was about the validation of your findings with additional external cohorts. Additional concerns were regarding discrepancies in reporting of your sample size in the manuscript and whether differences in the technical platforms used for the discovery and the validation data sets affect the validity of your results.

Please, submit a detailed rebuttal that shows where and how you have taken all comments and suggestions into consideration. If you do not agree with some of the reviewers’ comments or suggestions, please explain why. Your rebuttal will be critical in making a final decision on your manuscript. Please, note also that your revised version may enter a new round of review by the same or by different reviewers. Therefore, I cannot guarantee that your manuscript will eventually be accepted.

Reviewer 1 ·

Basic reporting

A few minor points mentioned in the 'General comments for the author'.

Experimental design

A few points mentioned in the 'General comments for the author' for improvement.

Validity of the findings

A few points mentioned in the 'General comments for the author' for improvement especially from 9 to 11.

Additional comments

In this manuscript, Gao et al. established a 5-circadian clock genes signature as a prognostic and predictive marker in TCGA and GSE72094 Lung adenocarcinoma (LUAD) datasets. Though the study is interesting and promising, the following points are recommended to be addressed to make this work more informative, confident, and reliable for the researchers and clinicians.

1. A couple of more lines in the background of the abstract should be mentioned. e.g., about the notoriousness of LUAD, what is circadian clock, and most importantly, the study's purpose.

2. Elaborate line 89, "Count data were normalized" adding more details, how and why.

3. Line 120, one-tailed or two-tailed Student's t-test?

4. Line 139, ‘of the’ is redundant.

5. Why did the authors do PCA, t-SNE analysis, and LASSO Cox regression analysis? Details will help the researchers who do not have the bioinformatics and statistics background e.g., the authors can frame the sentence like this: in order to find….we used PCA and showed…

6. Usually, in these types of studies, more than one validation cohort are used. Here only GSE72094 as the external validation cohort is used. Why specifically this cohort, and why only one being used? The signature of just 5 genes sometimes can lead to false-positive results physiologically; hence validation in more than one cohort will make this signature more reliable for the prognosis.

7. The results about the risk factors from table 1 and figure 4 are not discussed. These results need the discussion and what did the authors conclude for them.

8. Figure legends are not very informative. Please expand them and add more information to make them self-explanatory.

9. For Fig. 1B, the representation of the KM plot for all the five genes separately will be more informative for TCGA and GSE72094.

10. The heat map in Fig. 1C and its legend lacks the proper label and description. Is it mRNA expression?

11. For Fig. 1C, the representation of the differential expression of these genes between the tumor and the normal tissues, along with the heat map, will be more intuitive if the authors could show them as the box-plot or the violin-plot with individual points. At the moment, we do not know if the difference is significant or not because there are no p-values.

12. If possible, please look into other cancer types if this signature is impacting survival. This analysis is not necessary but will be valuable.

Reviewer 2 ·

Basic reporting

The title, abstract, introduction, methods, results and discussion are appropriate for the content of the text. Furthermore, the article is well constructed, the experiments are well conducted, and analysis is well performed. The figures are relevant, high quality, well labelled and described.

Experimental design

The experimental design is original and the research is within the scope of the journal.. Research question is well defined, relevant and meaningful. The methods are highly technical, ethical and logistical. Statistical methods are chosen correctly.

Validity of the findings

All underlying data have been provided in detail. The findings are meaningful. The conclusions are well stated and relevant to the research questions.

Additional comments

This paper investigates the function of expression of circadian genes in overall survival(OS) of lung adenocarcinoma(LUAD) by analyzing TCGA datasets. The authors identified a novel circadian gene signature by LASSO-penalized Cox regression analysis. Moreover, the authors validate their findings in a GEO dataset. To explore further about the related pathways, the authors demonstrate that circadian genes are functionally related to cell proliferation, gene damage repair, proteasomes, and immune and autoimmune diseases utilizing the GSEA method.

Editorial Criteria
BASIC REPORTING
The title, abstract, introduction, methods, results and discussion are appropriate for the content of the text. Furthermore, the article is well constructed, the experiments are well conducted, and analysis is well performed. The figures are relevant, high quality, well labelled and described.
EXPERIMENTAL DESIGN
The experimental design is original and the research is within the scope of the journal.. Research question is well defined, relevant and meaningful. The methods are highly technical, ethical and logistical. Statistical methods are chosen correctly.
VALIDITY OF THE FINDINGS
All underlying data have been provided in detail. The findings are meaningful. The conclusions are well stated and relevant to the research questions.

Overall, I think this paper is novel and will be of interest to the community of lung cancer genetics, especially LUAD research. The statistical part is valid and makes sense. The authors make it comprehensive by integrating analysis of multiple sources including GEO and TCGA. The main strengths of this paper is that it addresses an interesting and unexplored question, finds a novel discovery based on a carefully selected set of bioinformatic procedures. As such this article represents an excellent and elegant bioinformatics study which will almost certainly influence our thinking about the function of circadian genes in LUAD. Some of the weaknesses are the lack of in vitro or in vivo validation experiments. In general, the work is convincing except some major and minor comments below:


Major Comments:

I do see multiple platforms and normalization methods for LUAD samples in GDC, which includes HTSeq - Counts (sample size: 594), HTSeq - FPKM(sample size: 594) and HTSeq - FPKM-UQ(sample size: 594). Please explain why the sample size in the manuscript is 515 rather than 594. Please also explain why “HTSeq - FPKM” was chosen for analysis, rather than the other two?


I’m wondering if there are any ongoing clinical trials focusing on circadian clock genes in LUAD or other types of lung cancer? It will be very strong evidence for the significance of the current study if so.

I’m just wondering if the differences in sequencing platforms affects the results, since the discovery dataset(TCGA) is RNA-seq, and the validation dataset(GEO) is microarray. I’m wondering if a RNA-seq LUAD dataset as a validation dataset will produce more significant results.





Minor Comments:
Line 74: please replace “were extracted from the publicly available TCGA database” with “were retrieved from the NCI Genomic Data Commons”.

Line 63: please replace “using data extracted from The Cancer Genome Atlas (TCGA) database” with “using The Cancer Genome Atlas (TCGA) data obtained from the NCI Genomic Data Commons[add reference: https://www.nature.com/articles/s41588-021-00791-5 ]” .

It is great that a session of abbreviations was there to list all the abbreviations for the database names. I would recommend to also include abbreviations like LUAD, TCGA, TNM, GEO, DEG, LASSO, KEGG etc in that list.

All the gene names should be italic for all the gene names.

Figure 2E and 3E: the P value is not correctly formatted.

Annotated reviews are not available for download in order to protect the identity of reviewers who chose to remain anonymous.

---

## Round 0.2 · accepted · Accept

Thank you for doing a thorough job in addressing the reviewers' comments. As a result, your manuscript is much improved.

Reviewer 1 ·

Basic reporting

-

Experimental design

-

Validity of the findings

-

Additional comments

The authors addressed most of my points satisfactorily hence making this manuscript far better in every way. It can now be considered for publication!

Reviewer 2 ·

Basic reporting

The title, abstract, introduction, methods, results and discussion are appropriate for the content of the text. Furthermore, the article is well constructed, the experiments are well conducted, and analysis is well performed. The figures are relevant, high quality, well labelled and described.

Experimental design

The experimental design is original and the research is within the scope of the journal.. Research question is well defined, relevant and meaningful. The methods are highly technical, ethical and logistical. Statistical methods are chosen correctly.

Validity of the findings

All underlying data have been provided in detail. The findings are meaningful. The conclusions are well stated and relevant to the research questions.

Additional comments

This paper investigates the function of expression of ferroptosis-related genes in overall survival(OS) and prognosis of lung adenocarcinoma(LUAD) by analyzing TCGA datasets. The authors identified a ferroptosis-related gene signature by univariate Cox regression analysis. Moreover, the authors validate their findings in a GEO dataset. To explore further about the related pathways, the authors demonstrate that risk score was prominently enriched in ferroptosis processes. In short, this study identified the ferroptosis-related gene signature as a potential predictor for the prognosis of LUAD.

Overall, I think the corrections look good to me. All my comments have been answered in a proper manner. Please find the original comments and feedbacks below:

Editorial Criteria
BASIC REPORTING
The title, abstract, introduction, methods, results and discussion are appropriate for the content of the text. Furthermore, the article is well constructed, the experiments are well conducted, and analysis is well performed. The figures are relevant, high quality, well labelled and described.
EXPERIMENTAL DESIGN
The experimental design is original and the research is within the scope of the journal.. Research question is well defined, relevant and meaningful. The methods are highly technical, ethical and logistical. Statistical methods are chosen correctly.
VALIDITY OF THE FINDINGS
All underlying data have been provided in detail. The findings are meaningful. The conclusions are well stated and relevant to the research questions.

Overall, I think this paper is novel and will be of interest to the community of lung cancer genetics, especially LUAD research. The statistical part is valid and makes sense. The authors make it comprehensive by integrating analysis of multiple sources including GEO and TCGA. The main strengths of this paper is that it addresses an interesting and unexplored question, and finds a novel discovery based on a carefully selected set of bioinformatic procedures. As such this article represents an excellent and elegant bioinformatics study which will almost certainly influence our thinking about the function of ferroptosis-related genes in LUAD. Some of the weaknesses are the lack of in vitro or in vivo validation experiments. In general, the work is convincing except some major and minor comments below:


Major Comments:

The Methods of the Abstract are too simple, I would recommend expanding it and adding details like what datasets were used, what statistical methods were used to construct the prediction model, what pathway analysis methods were used. Some contents in the Results could be moved to the Methods.

Feedback: I did see the dataset and statistical methods were added in the method of the abstract. It looks good to me.


I do see multiple platforms and normalization methods for LUAD samples in GDC, which includes HTSeq - Counts (sample size: 594), HTSeq - FPKM(sample size: 594) and HTSeq - FPKM-UQ(sample size: 594). Please explain why the sample size in the manuscript is 528 rather than 594. Please also explain why “HTSeq - FPKM” was chosen for analysis, rather than the other two platforms?

Response:
There are 535 tumor samples (including 20 repetitive samples) and 59 normal tissue samples from 515 LUAD cases in TCGA. So there are only 515 cases with clinical data and 500 patients with OS data. We have described it in Line 79-83 and Line 136-145.

Feedback:
Thanks for adding the details. They look good.

I’m wondering if there are any ongoing clinical trials focusing on ferroptosis-related genes identified in this study in LUAD or other types of lung cancer? It will be very strong evidence for the significance of the current study if so.

Response:
We searched the website clinicaltrial.gov using “circadian” and “lung cancer”. There are only four ongoing or completed clinical trials. But all of them are about circadian rhythms and lung cancer. There is no clinical trial focusing on circadian clock genes in lung cancer.

Feedback:
Thanks for checking the clinical trials information. It’s good to know.

I’m just wondering if the differences in sequencing platforms affects the results, since the discovery dataset(TCGA) is RNA-seq, and the validation dataset(GEO) is microarray (GPL96 Affymetrix GeneChip Human Genome). I’m wondering if a RNA-seq LUAD dataset as a validation dataset will produce more significant results.

Response:
In this article, we use two independent data sets as the test set and the validation set. It is true that RNA-seq and microarray data are different from experimental methods, data acquisition, data normalization, and data standardization. That is why we did not combine the two datasets for analysis. Analyzing the two datasets separately avoids the bias caused by different experimental methods. And for LUAD, RNA-seq data with a large sample size cannot be found for verification in other databases.

Feedback:
Thanks for the explanation. It makes sense to me.



Minor Comments:
Line 63: : please replace “using data extracted from The Cancer Genome Atlas (TCGA) database” with “using The Cancer Genome Atlas (TCGA) data obtained from the NCI Genomic Data Commons” .https://www.nature.com/articles/s41588-021-00791-5
Answer: Agreed. Line 68.
Feedback: looks good now.

It is great that a session of abbreviations was there to list all the abbreviations for the database names. I would also recommend including abbreviations like LUAD, TCGA, GEO, DEG, GO, KEGG etc in that list.
Answer: It’s a good idea. If the editor needs it, we will list them.
Feedback: sounds good. Thanks!

All the gene names should be italic for all the gene names.
Answer: Revised.
Feedback: looks good. Thanks!

Figure 2E and 3E: the P value is not correctly formatted.
Answer: Revised.
Feedback: looks good. Thanks!

Annotated reviews are not available for download in order to protect the identity of reviewers who chose to remain anonymous.